# Breast Cancer Gone Viral? Review of Possible Role of Bovine Leukemia Virus in Breast Cancer, and Related Opportunities for Cancer Prevention

**DOI:** 10.3390/ijerph17010209

**Published:** 2019-12-27

**Authors:** Gertrude C. Buehring, Hannah M. Sans

**Affiliations:** 1School of Public Health, 16 Barker Hall, University of California, Berkeley, CA 94720, USA; 2School of Medicine, University of California, San Francisco, CA 94122, USA; hannah.sans@ucsf.edu

**Keywords:** breast cancer cause, bovine leukemia virus, environmental reservoir, primary prevention

## Abstract

This article is a literature review of research that explored the association of bovine leukemia virus (BLV) infection in humans with breast cancer. It summarizes and evaluates these publications. This review does not provide absolute proof that BLV is a cause of breast cancer, but, based on well-respected epidemiologic criteria for causation, it does suggest that BLV infection could be a breast cancer risk factor. Any expansion of the current understanding of breast cancer risk factors may increase possibilities to implement primary prevention strategies. The environmental role that BLV-infected cattle may play as a reservoir for infectious BLV offers possibilities for reducing or eliminating potential transmission of BLV from cattle to humans, and/or eliminating the reservoir.

## 1. Introduction

We all have heard that an ounce of prevention is worth a pound of cure. The problem, however, is that in order to prevent a disease we have to know what causes it, and that is not clear with breast cancer. Breast cancer is not caused by a single agent, but rather multiple single agents and/or various combinations of different agents. The following list summarizes, in two main categories, current estimates of the percentage of breast cancer cases in US women that are associated with various risk factors. Only 1% of breast cancers are in males. There is no definitive percentage for environmental chemicals because exposure is difficult to measure and multiple chemical factors are lumped into one category.
Genetic risk factors (cannot be altered)Genetics (mutated BRCA1, BRCA2, and/or other genes) [1] ~3%Environmental risk factors (non-genetic and possible to eliminate or avoid)Ionizing radiation (X-rays, Gamma rays, radon) [1] <2%Environmental chemicals in air, water, food, soil, dust, consumer products [2] ?%Long-term tobacco smoking [3] 37%


This literature review focuses on research that relates to bovine leukemia virus (BLV) infection of humans, and possible strategies to reduce human exposure to BLV. The sources of information are publications accessible in the PubMed Central database and on sources readily available online via Google.

## 2. Summary of Research on Relationship of BLV Infection to Breast Cancer

When Janice Miller and colleagues first discovered BLV in cattle in 1969, they were concerned that it could infect humans and cause disease [4]. Infected cattle had antibodies to BLV in their blood, which prompted early tests for anti-BLV antibodies in humans. However, eight serological surveys of humans for anti-BLV antibodies failed to give positive results. This prompted the oft-cited conclusion in 1996 by the National Animal Health Monitoring System (NAHMS): “BLV is not transmissible to humans and no human disease has ever been attributed to BLV” [5]. Despite this statement, many researchers have become interested in BLV and studied it from many perspectives, as reviewed by Gillet et al., [6]. Fortunately, the advent of the immunoblotting (western blotting) method prompted a retesting of human sera, and using this 100-times more sensitive technique, anti-BLV antibodies were clearly identified in the sera of 74% of 257 human subjects [7].

Another hurdle to overcome was the fact that production of human antibodies may not have been an exclusive indicator of infection with BLV because it could also represent an immune reaction to proteins in cooked beef and pasteurized dairy products consumed by the human population studied. This problem was remedied by the advent of PCR and DNA sequencing which allowed researchers to detect viral DNA, a more definitive marker of BLV presence. During the past five years several studies on BLV in breast tissues have been performed [8,9,10,11,12,13,14,15]. Their results are summarized in Table 1 below. 

The newer technology has also made it possible to gather enough data from different researchers to begin to apply the first four of “Hill’s epidemiologic criteria for causation of disease,” [16], and to evaluate BLV as a potential risk factor for breast cancer. The first criterion calls for a strong relationship between the potential risk factor being evaluated and the disease in question. In Table 1, the results (positive or negative) of the analysis methods listed in the third column from the left were used as the biomarker of exposure of the breast tissue to BLV, and a medical diagnosis of breast cancer was used as the marker of disease outcome. Potential disease/risk factor relationships are usually presented using two statistical terms: (1) “odds ratio” = the ratio of having the disease versus not having the disease when exposed to the agent in question; (2) probability = the probability of having the disease by chance. Table 1 (below) summarizes the results of eight studies exploring the relationship between BLV and breast cancer. Table 2 (below) compares the relative strength of the odds ratio of bovine leukemia virus infection with odds ratios of breast cancer risk factors that are well-established. 

**Hill Criterion #2 is consistency of results**, usually indicated by validation among different researchers, populations, and/or methods. Table 1 (above) indicates the results of BLV presence in breast epithelium in 5 out of 8 populations globally. 

**Hill Criterion #3**: **Specificity of association of causal agent and disease**. PCR oriented data requires a very strict design of the primer sequences, which targets the specific region of the DNA genome for amplification. The primers used to detect BLV in breast tissues were checked for specificity on a website called Standard Nucleotide BLAST (Basic Alignment Search Tool), a noncommercial free site offered by the US Department of Health and Human Services. When a short primer DNA sequence is submitted to the site, it will indicate how similar that sequence is to the greater than 162 million sequences deposited in this reference base. The specificity of the primer sequences is assured if there is close relationship to the sequence of DNA the researcher desires to amplify (low E value), and also an extremely low relationship to the human genome (high E value). The specificity values are all numerical estimates of any similarity being by chance and therefore can be easily compared. The first five studies in Table 1 all used the same primers, which had a close relationship to BLV sequences and an extremely low relationship to the human genome [7,8,9,10,11].

As an additional guarantee of specificity for BLV, these same primers were tested on cell lines infected with other retroviruses closely related to BLV, and also with proven cancer viruses, to make sure there was no positive reaction. DNA amplification occurred only in cell lines infected with BLV and did not occur in cell lines infected with closely related retroviruses, endogenous retroviruses, or viruses known to cause other cancers in humans [17].

**Criterion #4**: **The exposure must precede the outcome.** I.e., the causative agent must be present in the tissue in question before the cancer develops. If it first appears in the tissue after the cancer has developed it could be interpreted as the result, rather than the cause of the cancer. This temporal relationship is usually difficult to demonstrate because it requires the collection of two specimens from the same patient, each removed from the patient years apart. We were lucky to obtain one normal breast specimen at the time of the first surgery and one normal or malignant specimen from the same breast 3–10 years later from 31 Australian women. Of the women who were BLV+ at the first surgery (normal tissue), 60.4% were BLV+ on the second surgery (breast cancer). Of the women who were BLV− on the first surgery (normal tissue), only 14.6% were BLV positive on the second surgery (breast cancer) [9].

**Criterion #5**: **The cancer appears more frequently in humans exposed to higher, compared to lower, doses of the causative agent**. There are very few examples of this because breast cancer patients do not usually have quantitative information about the risk factors that she or he might have been exposed to. However, biological gradients have been observed for breast cancer when the approximate frequency of tobacco smoking or ionizing radiation was known. Inferences can also be made from data on large populations, as in the Figure 1 below, showing similarity in geographical distribution of the darker shades of blue which indicate higher rates of milk consumption and breast cancer incidence in the left and right map, respectively [18]. Interestingly, geographical areas with large numbers of lactose intolerant people (Central Africa and East Asia), have relatively low milk consumption and low breast cancer incidence, compared with other areas with a more “Western” culture.

**Criterion #6: Plausibility**. Does the causative agent mechanism of action fit with disease and with similar agent/disease relationships? Six types of human cancer (uterine cervix, liver, head/neck, lymphoma, Kaposi sarcoma, Merkel cell carcinoma) have already been proven to be caused by a virus, so the debut of a new virus/cancer relationship should not be surprising. One possible mechanism of carcinogenic action of BLV and its close relative HTLV (human T-cell leukemia virus) is inhibition of cellular DNA repair [19], consistent with how certain mutated genes (BRCA1 and BRCA2) contribute to the development of breast cancer.

**Criterion #7**: **Coherence with other data from animal and in vitro models.** Cattle and sheep infected with BLV develop leukemia and lymphomas. Mammary epithelial cells of BLV-infected cows become infected [20] and their nursing calves usually become BLV infected, likely through BLV infected cells in their milk. Mammary cancer in cows has not been reported, maybe because many are slaughtered for the market at a relatively young age. Could the physical pressure of a nursing calf or a milking machine result in the shedding of mature BLV-infected cells into the milk before they have had a chance to develop into malignant cells? This has not yet been explored. 

**Criterion #8**: **Experiments support the hypothesis.** BLV-infected cells pass the virus to noninfected cells by cell-cell contact in vitro (unpublished data). BLV-infected cells injected into BLV-free animals result in active infection [21].

## 3. Is Evidence That BLV Might Be a Risk Factor for Breast Cancer Now Strong Enough to Consider Primary Prevention Strategies to Lower BLV Infection Rate in Humans?

Collectively, the high level with which BLV is associated with breast cancer and meets the expectations of the Bradford Hill criteria should get more attention in future approaches to combat breast cancer. Currently, the main approach is early detection and treatment to prevent mortality. This is, of course, important and should not be discontinued. But why not pay more attention to primary prevention, i.e., protecting humans from being exposed to risk factors for breast cancer so that they never develop breast cancer in the first place? For breast cancers caused by BLV, several pathways are possible:
1.Determine how humans become infected with BLV so those routes of transmission can be intercepted. Among cattle, BLV is transmitted via milk, blood (biting insects, contaminated veterinary/farming equipment, in utero) [22] so it is likely humans are infected through these same bodily fluids. Estimated prevalence of BLV in US cattle herds is 84% of dairy herds and 38% of beef herds [22]. However, BLV is inactivated by pasteurization of milk and thorough cooking of beef, so educating people not to drink raw milk products or eat raw or extremely rare beef would protect them against this route of transmission.


Another possible route of transmission is human-to-human via blood or milk. Currently, blood banks don’t test for BLV when they screen human blood donations. A recent study of human blood indicated 45% of subjects who volunteered had leukocytes infected with BLV [23], suggesting that blood borne infection from other humans might be possible, however, no research has been done to prove this. Mother-to-child transmission via nursing and blood exposure during the birth process is another possible human-to-human transmission method which needs to be explored.
2.Eliminate BLV infection from US cattle, a likely main source of BLV infection.
Eliminate farm/ranch practices that spread BLV among animals (e.g., using the same veterinary and agricultural equipment on multiple animals without decontamination between individual animals).Submit blood specimens from each animal to a veterinary testing laboratory to test for BLV. Then completely separate the pastures, barns, and all equipment used for BLV-positive from those used for BLV-negative animals.Figure 2. BLV positive and negative cattle separated by two fences [24].Create the future herd only from negative animals. This strategy has already achieved successful eradication of BLV in cattle in Australia/New Zealand and 19 nations in Europe [22].An anonymous pilot survey was sent to all registered owners of California beef and dairy farms in hopes of gathering information about their farm practices, their knowledge of BLV and other cattle viruses, and their attitudes about testing and eliminating BLV from their herds. Interestingly, the response rate was 1/44(2%) for dairy farmers and 16/42 (38%) for beef farmers. This inversely corresponds to the known rate of BLV infection in dairy herds (84%) versus beef herds (38%) and suggests that dairy farmers may already be aware of the high rate of BLV infection in dairy herds and may be hesitant about responding to the questionnaire. The survey also indicated that the majority of the beef ranchers who responded were members of an agricultural organization, which might be a persuasive vehicle of gaining the cooperation of ranchers/farmers in any proposed BLV elimination program.
3.The ideal solution for prevention of most infectious diseases is an effective vaccine which, in the case of BLV infection, could be used to vaccinate both cattle and humans. Initial attempts to develop an effective vaccine to protect cattle from BLV infection resulted in antibody formation Bovine leukemia virus associated with mammary epithelial cell proliferation in Argentinian women but did not protect against a challenge with infectious BLV injected later [21]. This suggests that the only effective vaccine may have to be a live vaccine, which is much more complicated to develop. However, new strategies for an effective vaccine look promising. If more funding were available to support further research in this area, scientists would likely be more interested in live BLV vaccination development.


## 4. Conclusions

There are already multiple risk factors for developing human breast cancer, and bovine leukemia virus (BLV) may be another one to add to the list. Six of the eight studies that examined breast cancer tissues or DNA found BLV in breast tissues, which suggests strongly that BLV does infect humans, and breasts can be targets of infection. Four of the five studies that were able to obtain normal breast tissue from donors without breast cancer performed a comparative statistical analysis and found relatively high odds ratios, ranging 2.73–5.59, with an average of 4.01. Although this does not definitively mean that BLV can cause breast cancer, it is strong support for the idea that BLV may be a risk factor. The elimination of the source of BLV and/or its transmission may help to reduce breast cancer incidence and mortality in both women and men. Most BLV around the world is housed in the environment, namely the agricultural pastures in which our large domestic animals live until they become part of our diet. These pastures are frequently adjacent to open land inhabited by wild animals that could potentially become infected. Unlike other viruses already proven to cause human cancers by human-to-human transmission (liver cancer, cervical cancer, Kaposi’s sarcoma, Burkitt’s lymphoma), BLV crosses species readily and has been shown to infect a variety of wild animals (yaks, buffalo, capybara) [25] in addition to domestic sheep and goats. These uncontrolled infections increase the reservoir of BLV and ultimately the probability of human infection. Because of the expense, many beef cattle ranchers and dairy farmers are reluctant to attempt eradication of BLV from their herds by testing and separating positive from negative animals and creating a future herd from the negative animals. Therefore, any campaign to eliminate BLV infection in cattle for the sake of human health should also emphasize the potential benefits of BLV eradication to ranchers and farmers, e.g., BLV-negative cows have higher level of milk production and greater resistance to other microorganisms that lower the general health and marketability of the cattle. Therefore, eradicating the zoonotic carcinogenic risks within this agricultural environment could have huge payoffs for both human health as well as the agricultural industry. 

## Figures and Tables

**Figure 1 ijerph-17-00209-f001:**
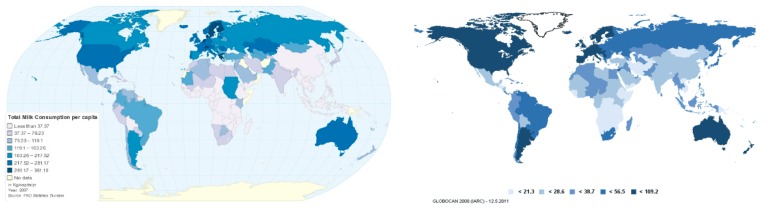
Global geographical comparison of milk consumption with breast cancer incidence. Left: Estimated milk consumption (2008) [17]; Right: Estimated breast cancer incidence (2008).

**Figure 2 ijerph-17-00209-f002:**
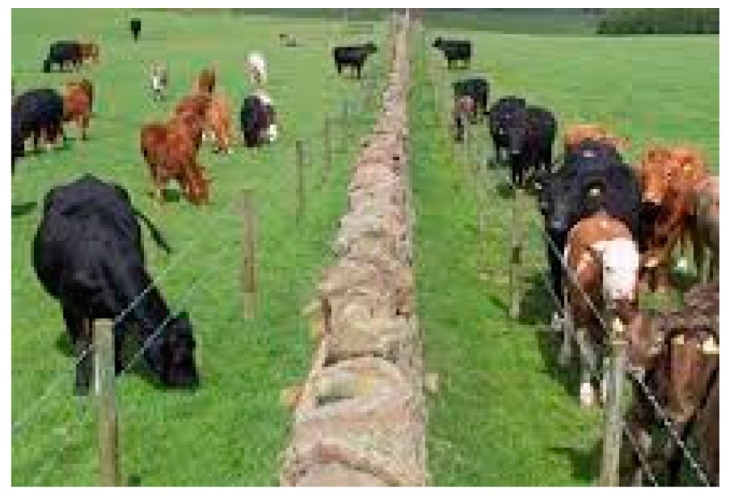
BLV positive and negative cattle separated by two fences.

**Table 1 ijerph-17-00209-t001:** Hill Criterion #1: There should be a strong relationship between causative agent and disease in question.

Source of Specimens/[Reference]	Sample Number/Specimen Type	Analysis Methods	Normal Tissue	Cancer Tissue	Odds ratio (Risk) (95% Confidence Internal)	Probability Happening by Chance *
1. Cooperative Human Tissue Network (Southern and Eastern Division [8]	*n* = 239/FFPE	IS-PCR	30/104 (29%)	67/114 (59%)	3.07 (1.66–5.69)	*p* < 0.001
2. Douglass, Hanley, Moir Pathology, Macquerie Park, NSW, Australia [9]	*n* = 96/FFPE	IS-PCR	19/46 (41%)	40/50 (80%)	4.72 (1.71–13.05)	*p* < 0.003
3. MD Anderson Cancer Center, Houston, TX [10]	*n* = 166/FFPE	IS-PCR	20/105 (19%)	35/61 (57%)	5.59 (2.76–11.30)	*p* < 0.0001
4. Argentina Buenos Aires/Tindal [11]	*n* = 85 FFPE	IS-PCR	0	19/85 (23%)	Analysis not possible; no normal tissue available	
5. Columbia Bogota [12]	*n* = 53	PCR	24/53 (45%)	20/53 (36%)	Analysis not performed by authors; normal tissue had higher % positive	
6. Brazil, Rio Grande do Sul [13]	*n* = 144 FFPE	IS-PCR NL-PCR	10/72 (14%)	22/72 (31%)	2.73 (1.18–6.29)	*p* < 0.027
7. USA, Mexico, Vietnam [14]	*n* = 51 USA NCI DNA sequence data	NGS	0	0/51 (0%)	Analysis not possible; no normal breast tissue sequences available	
8. China [15]	*n* = 91 Breast tissue Blood serum	RT-PCR? ELISA	0	0/91 (0%)	Analysis not possible; no normal breast tissue available	

* Calculations based on comparison of malignant and normal samples. Abbreviations: Con = conventional; ELISA = enzyme-linked immunosorbent assay; FFPE = formalin fixed paraffin embedded; IS = in situ; M = malignant; *N* = nonmalignant; NCI = National Cancer Institute; NL = normal liquid PCR; NGS = next generation sequencing; PCR = polymerase chain reaction; RT = real-time pPCR.

**Table 2 ijerph-17-00209-t002:** Hill Criterion #1: relative strengths of breast cancer risk factors.

Risk Factor	Referrant	High Risk Group	Risk *
Years on hormone replacement	None	5 years	1.3
Age at menarche	>15 years	<12 years	1.3
Age at natural menopause	<45	≥55 years	1.4
Years on oral contraceptives	none	>12 years	1.4
Parity	≥5	Nulliparous	1.4
Postmenopausal BMI	<22.9	>30.7	1.6
Age at first full-term pregnancy	<25 years	>35 years	1.8
First degree relative with BC	None	One	2.0
Mother/sister with breast cancer	Not present	Present	3.6
BLV in breast epithelium	Not present	Present	4.0 ^#^
BRCA1 and BRCA2 genes	Not mutated	Mutated	4.7
Ionizing radiation	None	High dose	5.2

* risk or potential is given as odds ratio; # average of the 4 studies that calculated an odds ratio. Risk measured as odds ratio or relative risk; BC = breast cancer; BLV = bovine leukemia virus; BMI = body mass index

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
