# Peer review of "Breast Cancer Gone Viral? Review of Possible Role of Bovine Leukemia Virus in Breast Cancer, and Related Opportunities for Cancer Prevention"

_ijerph, 2019, doi:10.3390/ijerph17010209_

Round 1
Reviewer 1 Report
Into the present form, the manuscript could be accepted. I appreciate a lot that the authors write: “Although this does not definitively mean that BLV can cause breast cancer, it is strong support for the idea that BLV may be a risk factor”and I think that could be appropriate to add into the abstract that this is not a demonstration but an idea supported by different signs.
Author Response
We have revised the abstract as the reviewer requested: making the language more modest. We have eliminated language that sounds like it is claiming that there is strong evidence that bovine leukemia virus (BLV) may be a cause breast cancer. We have replaced this thought with a statement that we have summarized epidemiologic evidence that human infection with BLV might be a risk factor for developing breast cancer.
Breast cancer gone viral? Review of possible role of bovine leukemia virus role in causation breast cancer, and related opportunities for cancer prevention
Gertrude C. Buehring1 and Hannah M. Sans2
1 Affiliation: School of Public Health, University of California, Berkeley, buehring@berkeley.edu
2 Affiliation: School of Medicine, University of California, San Francisco, hannah.sans@ucsf.edu
* Correspondence: buehring@berkeley.edu;
Received: date; Accepted: date; Published: date
Abstract: This article is a systematic literature review of research that has explored explores the role association of bovine leukemia virus (BLV) infection of humans as a cause of with breast cancer. It summarizes and evaluates these research publications. that support causation by BLV and the few that don’t. This review does not provide absolute proof that BLV is a cause of breast cancer, but, based on well respected epidemiologic criteria for causation, it does suggest that BLV infection could be a breast cancer risk factor. Any expansion of the current With this clearer understanding of breast cancer risk factors causation, it may be possible increase possibilities to implement primary prevention strategies. of as many as 50% of all new breast cancer cases in the USA. The environmental role that of BLV infected on of cattle may play as a reservoir for infectious BLV offers possibilities for reducing or eliminating potential transmission of BLV from cattle to humans. from the cattle reservoir and/or eliminating that the reservoir.
Reviewer 2 Report
Introduction is scant and doesn’t focus in a precise manner on the topic of the manuscript. A huge number of article dissect general aspect concerning etiology and pathogenesis of breast cancer. Useful information are reported in the following manuscript and therein refs (https://doi.org/10.3389/fendo.2018.00492; https://www.nature.com/articles/s41598-019-39834-7).
I suggest to the authors to refer only to literature data reported in PubMed Central or article published on impacted journals.
A stringent revision of English form should be performed.
A deeper description of BLV should be included. What is its mechanism of action? What is its tropism?
Please clarify better the study design of evidence you referred to. Is there a comparison between infected and non infected breast cancer patients? Did the virus affect the prognosis?
Lines 96-102 are difficult to understand.
Author Response
Into the present form, the manuscript could be accepted. I appreciate a lot that the authors write: “Although this does not definitively mean that BLV can cause breast cancer, it is strong support for the idea that BLV may be a risk factor”and I think that could be appropriate to add into the abstract that this is not a demonstration but an idea supported by different signs.
Below is my remodeled abstract to try to satisfy the reviewer’s comments above.
This article is a literature review of research that explored the role of bovine leukemia virus (BLV) as a possible cause of breast cancer. It summarizes and evaluates these publications. This review does not provide absolute proof that BLV is a cause of breast cancer, but, based on well respected epidemiologic criteria for causation, it does suggest that BLV infection could be a risk factor for developing breast cancer. Given the current minimal understanding about breast cancer causation, any reasonable information on this subject could help in planning future primary prevention strategies. The environmental role that BLV infection of cattle may play as a reservoir for infectious BLV offers possibilities for reducing transmission to humans from the cattle reservoir and/or eliminating that reservoir.
Round 2
Reviewer 2 Report
I accept the manuscript in the present form
This manuscript is a resubmission of an earlier submission. The following is a list of the peer review reports and author responses from that submission.
Round 1
Reviewer 1 Report
Line 8-9 These studies show association, not causation.
Line 9-10 If 100% of the association is causal in nature and 100% of this causal pathway can be prevented, then this statement could be true. However, it seems misleading to only mention the most extreme favorable outcome.
Line 9 What environmental role? BLV transmission among cattle is usually by direct contact or sometimes vectorborne or bloodborne. People would probably be exposed in the same way, but almost certainly not through environmental exposure. Unless the very broad definition of environment is being used in which everything that is non-human is considered environmental exposure.
Line 18-26. Again – it is very incorrect to claim that associations found on observational studies are 100% causal in nature. It is very premature to claim that 50% of human breast cancer is caused by BLV.
What happened to criteria #1? And I assume there was an explanation and reference for these Hill’s causal criteria. Maybe the table copied over the top of this. This happens a lot in Word.
Line 88 to 96 Was it shown that the rate of cancer among PCR-positive women was higher than among those that had previously been found to be BLV-infected as compared to those not-BLV infected? If BLV and BC are non-causally associated, they you would expect that some BLV would happen before BC and visa versa.
Line 97-103 Many western European countries eradicated BLV 20 or more years ago. Why are their rates of breast cancer (BC) higher than U.S. rates? Are BC rates higher in those with farm exposure? Lower in lactose-intolerant people? A good review article should not sift through the available information and only report that evidence that supports the author’s conclusion.
Line 117 I’d say “maybe” not probably. Looking at percentages, it is true that dairy cattle are usually slaughtered at a very old age. But millions each year are slaughtered at a very old age that would have given plenty of time for BC to develop.
Line 132. Biting flies, direct contact, in utero,
Line 145 Brucellosis vaccine no longer required in most states.
Line 170 to 171. This sentence is unsupported – as described above and below.
Line 175-176. No wildlife reservoirs have ever been reported.
A “systematic review” would entail a higher level of objectivity in the review of the literature without bias towards one particular conclusion. This review is not. 8 of 20 references cited were her own papers, while only including 4 papers (Citations 11-14) contradicted her hypothesis, totaling 3-4 sentences (line 45-52).
Dismissal #1:
Within this passage (Line 45-52), it was noted that the lack of detection of BLV via nest-generation sequencing does not “usually” integrate into the host genome (citing a review on retroviruses). This claim is unfounded for three reasons: (1) there is no evidence that this is the case for BLV (in cows or humans) and (2) NGS Library prep often does include PCR whole genome amplification, while bridge amplification of hybridized DNA fragments occurs during the sequencing run within the flow cell. (See page 5 Figure D Illumina’s Overview). (3) NGS sequencing is extremely sensitive in its ability to detect low abundant DNA molecules due to the massively parallel processes taking place leading to hundreds of millions of reads per run.
Dismissal #2:
Within the Chinese study which demonstrated that no BLV seropositivity within Chinese women, it is noted that the kit used was indicated for bovine immunoglobulin, however there is more than sufficient evidence in the literature that secondary antibodies are polyclonal in origin and do typically cross-react to primary antibodies from multiple species, especially within mammals. Additionally, Reference 5 which shows that human sera is reactive to recombinant p24 via western blot, used protein G beads to conjugate HRP rather than secondary antibodies which directly contradicts (“cherry picks”) the logic used to dismiss claims made by Zhang et. al (Chinese antibody breast cancer paper; Citation [5])
Interpretation of Table 1:
At no point are the Sensitivity/specificity issues regarding FFPE in situ PCR and nested PCR included in the body of this manuscript as an objective discussion to qualify the findings summarized in Table 1. Both methods require stringent negative controls due to their proclivity to produce false positives. Additionally, the studies listed in support of this hypothesis (#6-8) cite the authors own work. To enumerate the claims and shortcomings of these papers:
[6] IS-PCR (figure 1), the negative control is “background staining”, excluding the presence of the primers used to “detect” BLV DNA. This negative control is not sufficient to conclude that positive staining is specific to BLV DNA due to the lack of evidence that these primers are not cross-reacting to formalin-fixed human DNA or simply produce artifactual staining.
[7] IS-PCR and IS-hybridization (ISH) on FFPE (Figure 1). This experiment lacks controls (both positive, FLK cell line smear (mentioned) and negative, intact IS-PCR conditions, containing primers, on BLV-negative human tissue). When compared to ISH detection of HPV (Figure 3) These details of the how the negative control was performed (inclusion of ISH probe) are omitted. Given that this is an established method for HPV in cervical tissues, ISH probes were most likely included. Dr. Nuovo’s textbook was cited as the methods used for HVP ISH, not a specific primary publication.
[8] in situ-PCR using a hybridization probe on FFPE sections (Figure 1). This experiment is also missing the critical negative control which has the presence of the in-situ probe on BLV negative tissue. This is noted in the figure legend that the presence of the probe leads to “artifacts inherent in some tissues”.
In conclusion, the work summarized in this “systematic review” reflect more of a self-fulfilling scientific review of one viewpoint based on less-than robust studies. Contradictory logic is used to support the authors position. This paper is not really a comprehensive and objective summary of the connection of BLV and human breast cancer. The suggestion of causation is premature and unsubstantiated. For instance, listed in the abstract that up to 50% of all new US breast cancer cases may be prevented in the absence of BLV’s existence without mention or investigation into the mode of transmission or mechanism of carcinogenesis.
Author Response
Thank you for your perceptive comments. Our responses are in italics following each of the reviewer's comments.
1- The authors sustain that: "the proven mechanism of carcinogenic action of BLV is inhibition of cellular DNA repair" but different cancers have one mutation that involves the DNA repair system so, why the BLV infection transforms, in particular, the breast tissue?
It is quite clear that the wording was incorrect and we should have said “one proven mechanism of potentially carcinogenic action of BLV….” Rather than “the proven method….” There are multiple mechanisms of carcinogenesis that BLV and other carcinogens are capable of. We tested only one mechanism of carcinogenic action, the inhibition of DNA repair of damage caused by metabolic intracellular oxidative damage, and we found that the Tax protein of both BLV and its close relative HTLV (human T-cell leukemia virus) inhibited the base-excision repair of oxidative damage. This mechanism might contribute to cell transformation by these retroviruses. (J. National Cancer Institute 199;, 91:933-42). We did not give full details on the research just described because it is available in the JNCI journal, and a large part of the audience who may read the IJERPH are public health professionals who are not molecular biologists. Our experiments were in vitro and utilized over 30 different cell lines from different species and tissue types. We focused primarily on breast cancer because that is the focus of our research. We did not test for the presence of BLV in other human solid tissues, but hope that other scientists will do that, so the scientific public will know the extent of BLV infection in humans.
2- Why does this virus never cause breast cancer in animals?
We do not know the answer to your question but would guess that for cattle it’s because cows have co-factors that are very different from those of humans: e.g. shorter life span, sometimes confined and protected living environment, vegetarian diet, etc. We do know, however, that BLV infects the mammary epithelial cells of cattle and these cells exfoliate and are plentiful in the milk that cattle produce (Buehring GC, Kramme PK, Schultz RD, Evidence of bovine leukemia virus in the mammary epithelial cells of infected cows. Lab. Investigation 71:359-365, 1994). This information has been added in the section titled Criteria #7. Maybe the physical pressure of constant milking of cows causes the cells to be shed before they get a chance to develop into cancer cells? Mammary cancer does occur in a variety of domestic animals: dogs, cats. and domesticated lions, tigers in the zoo. However, we could not find any published articles on research to determine if the mammary cancer tissue of these animals is infected with BLV, so one cannot conclude that BLV never causes breast cancer in animals.
3-Is it possible that breast cancer arises not directly by BLV infection but is only a con-cause or simply a consequence of an inflammatory stage caused by BLV? In this last case, if the authors consider another viral infection (a virus that act in the same way), can they obtain a correlation between breast cancer and viral infection?
The initial steps of breast cancer can probably arise in a variety of ways and inflammation sounds quite logical. We have not looked at any other viruses for their potential to cause breast cancer, mainly because we are focusing on BLV and do not have the personnel or funding to study a lot of other viruses, but there are others who are focusing on other retroviruses. Yes, it is very likely that other co-factors are required for BLV to play a role in any kind of cancer. Virtually every virus and other infectious agents are affected by other factors that determine whether or not a particular human will develop the disease the infectious agent causes. Most humans that become infected by any virus or other infectious agent, don’t develop the particular disease, yet the infectious agent is still considered the cause of the disease. It’s called the “iceberg concept of disease.” For cancer viruses, the effectiveness of the immune system, genetics, gender, age, reproductive history, etc. are very important co-factors. An excellent example is HPV (human papilloma virus) which is now widely accepted as the cause of cancer of the uterine cervix. It is a common infection of women. However, in the USA, for example, only 1% of all women with HPV detected in their cervical tissue actually develop cervical cancer during their lifetime. This indicates strongly that there are important co-factors involved. Human T-cell leukemia virus (HTLV), which is closely related to BLV, also inhibits base excision repair of oxidative damage of DNA repair. Although HTLV has not been studied much for a possible role in breast cancer, a recent study of 610 breast cancer patients, showed no association of breast cancer with HTLV infection (Asian Pac J Cancer Prev. 2019 Jun 1;20(6):1909-1912. doi: 10.31557/APJCP.2019.20.6.1909).
4-In Countries with a high number of grazing cows and a high meat consumption (for example Argentina), there is not an high breast cancer incidence (23%). In Countries with less industrialization, where people could drink not pasteurized milk, the high milk consumption does not correspond to an high breast cancer incidence (for example the African regions are not the same) as shown in fig at lanes 115-116). How do the authors explain this Please note that this figure have not a number and modify.
The 23% does not represent the incidence of breast cancer in Argentina. It represents the frequency of BLV in the breast cancer tissues from Argentina. Other factors in Argentina could possibly be responsible for initiating the development of breast cancer. BLV is not the only factor associated with breast cancer. The maps of the world in Figure 1, indicate a much lower incidence of breast cancer in Africa and a few other parts of the world than in countries such as Argentina, with a western style diet. Most people with African and/or east Asian ancestry are lactose intolerant and so do not drink much milk. Also in many areas of central Africa, the cattle are breeds traditional to those areas and usually BLV negative. Whereas other parts of Africa with a largely Western culture have breeds of cattle originating in Europe and are more frequently infected with BLV.
Thank you for your suggestion to number the Figure. We have done that.
Reviewer 2 Report
In this review, the authors try to highlight a possible correlation between BLV infection and breast cancer. They collect different data in literature and conclude that the BLV could be responsible of the 50% of breast cancer in women. Even if original, this study is not really convincing. To this purpose, I have different questions.
1- The authors sustain that: "the proven mechanism of carcinogenic action of BLV is inhibition of cellular DNA repair" but different cancers have one mutation that involves the DNA repair system so, why the BLV infection transforms, in particular, the breast tissue?
2- Why does this virus never cause breast cancer in animals?
3-Is it possible that breast cancer arises not directly by BLV infection but is only a con-cause or simply a consequence of an inflammatory stage caused by BLV? In this last case, if the authors consider another viral infection (a virus that act in the same way), can they obtain a correlation between breast cancer and viral infection?
4-In Countries with a high number of grazing cows and a high meat consumption (for example Argentina), there is not an high breast cancer incidence (23%). In Countries with less industrialization, where people could drink not pasteurized milk, the high milk consumption does not correspond to an high breast cancer incidence (for example the African regions are not the same) as shown in fig at lanes 115-116). How do the authors explain this Please note that this figure have not a number and modify.
Author Response
Our responses are in italics following each of the reviewer's comments.
Comment 1. Line 8-9 These studies show association, not causation.
Thank you for your thorough review and perceptive comments. Yes, we agree it is premature to claim that the associations of BLV and breast cancer found could be causal. Far more evidence would be needed to make this idea fit convincingly with causation. The manuscript was meant to be a summary of the research that has been done so far to determine if humans are really infected with BLV and if so, is its association with breast cancer statistically significant.
We have reworded the entire manuscript text to eliminate the words “cause” and “causation” in most places and use only “association.” We have replaced the word “cause” with “risk factor” in a few sentences where “risk factor” instead of “associated”, was needed for clarity or for context. In the field of epidemiology, a “risk factor” is defined as a variable associated with an increased risk of disease or infection. It does not mean the same thing as “cause.”
Line 9-10 If 100% of the association is causal in nature and 100% of this causal pathway can be prevented, then this statement could be true. However, it seems misleading to only mention the most extreme favorable outcome.
Virtually every virus and other infectious agents are affected by other factors that determine whether or not a particular human will develop the disease the infectious agent causes. Most humans that become infected by any virus or other infectious agent, don’t develop the particular disease, yet the infectious agent is still considered the cause of the disease. It’s called the “iceberg concept of disease.” For cancer viruses, the effectiveness of the immune system, genetics, gender, age, reproductive history, etc. are very important co-factors. An excellent example is HPV (human papilloma virus) which is now widely accepted as the cause of cancer of the uterine cervix. It is a common infection of women. However, in the USA, for example, only 1% of all women with HPV detected in their cervical tissue actually develop cervical cancer during their lifetime. This indicates strongly that there are important co-factors involved.
We revised the Abstract to eliminate all words related to causation, e.g. “cause,“causative,” and “caused” and make the theme only about “association” or a risk factor.
Line 9 What environmental role? BLV transmission among cattle is usually by direct contact or sometimes vectorborne or bloodborne. People would probably be exposed in the same way, but almost certainly not through environmental exposure. Unless the very broad definition of environment is being used in which everything that is non-human is considered environmental exposure.
Risk factors for cancer are often segregated into two categories for epidemiology, broad statistics, and for reporting to the public: Category 1 =“Genetic,” which cannot be altered and Category 2 = “Environmental,” exposures which can be altered through avoidance or elimination of the factor, e,g, diet, air, water, alcohol, tobacco, viruses and other infectious agents, etc. This would be called a broad definition but it is not related to species. We revised the text in the Introduction to present various well-known “risk factors” for breast cancer in either the Genetic or Environmental category. The formatting of the known risk factors as Genetic or Environmental is taken directly from reference 1 and 2. We eliminated any mention of BLV that we had previously put in this part of the text.
Line 18-26. Again – it is very incorrect to claim that associations found on observational studies are 100% causal in nature. It is very premature to claim that 50% of human breast cancer is caused by BLV.
Yes, we agree it is premature. That sentence has been deleted.
What happened to criteria #1? And I assume there was an explanation and reference for these Hill’s causal criteria. Maybe the table copied over the top of this. This happens a lot in Word.
Criteria #1 is in Table #1. We are assuming the editorial staff will want the Table numbering to take precedence.
The reference for the Austin Bradford Hill criteria was listed as #14 in the reference list. Numerous criteria sets have been developed by various scientists (Henle-Koch’s postulates, Pagano, Lilienfeld, Sartwell, Fredericks and Relman, US Surgeon General Report). All except the Koch’s postulates are based primarily on epidemiology/statistics. The Hill (Austin Bradford Hill) criteria are currently the most widely used and respected of all disease causation criteria for diseases of humans. But Hill, himself, warned that the criteria were a conceptual guide, and not a set of stringent requirements that all had to be satisfied perfectly.
Line 88 to 96 Was it shown that the rate of cancer among PCR-positive women was higher than among those that had previously been found to be BLV-infected as compared to those not-BLV infected? If BLV and BC are non-causally associated, they you would expect that some BLV would happen before BC and visa versa.
Yes, thank for this insight. Of the women who were BLV+ at the first surgery (normal) (60.4% were BLV+ on the second surgery (breast cancer). Of the women who were BLV-neg. on the first surgery(normal tissue), only 14.6%.were BLV positive on the second surgery (breast cancer). This information has been added to the text.
Line 97-103 Many western European countries eradicated BLV 20 or more years ago. Why are their rates of breast cancer (BC) higher than U.S. rates?
It is estimated that it takes 20-50 years for breast cancer to develop after exposure to the carcinogenic agent, so any reduction in breast incidence in Europe would just be beginning. Also, there are many factors that contribute to breast cancer incidence besides the initial factor that starts the process. Some experts hypothesize that breast cancer incidence is increasing in Europe because the European socioeconomic level has increased. More women are employed and having children at an older age, which is a breast cancer risk factor. Also, if Europe is offering more affordable mammography to diagnose breast cancer earlier, there would result in more cases on record. The European population is getting richer and living longer, which increases the risk of breast cancer in both women and men and makes it more difficult to analyze what effect the reduction in BLV exposure would have, unless the breast cancer specimens were all tested for BLV. BLV presence in breast cancer specimens so far, has only been done in populations from the USA, Argentina, and Brazil.
Are BC rates higher in those with farm exposure? (Don’t know of any statistics on farm exposure per se) Lower in lactose-intolerant people? (Yes) See lines 140-143, just above the maps of geographic distribution of breast cancer and milk consumption.
A good review article should not sift through the available information and only report that evidence that supports the author’s conclusion.
Yes, we agree. Your perceptive comments have helped us to realize how much information was missing from this manuscript and needed to be provided to enhance the readers’ understanding of basic epidemiology and cancers caused by viruses. The manuscript is now much longer and hopefully more informative.
BLV was eradicated from the first European countries in 2003 and a few later countries in Europe in 2011.
That was only 16 and 8 years ago respectively. That is not enough time for most breast cancers to develop, given the estimated latency period between initiation of the cancer process and the appearance/diagnosis of cancer.
Line 117 I’d say “maybe” not probably. Looking at percentages, it is true that dairy cattle are usually slaughtered at a very old age. But millions each year are slaughtered at a very old age that would have given plenty of time for BC to develop.
Good suggestion! We couldn’t find the word “probably” at line 117, but found it at Line 127 in the correct context and it has been changed to “maybe.”
Line 132. Biting flies, direct contact, in utero,
We added these to the text.
Line 145 Brucellosis vaccine no longer required in most states.
Thanks for the information. The text about brucellosis has been omitted and replaced by “contaminated veterinary/farming equipment” (see lines 200-203)
Line 170 to 171. This sentence is unsupported – as described above and below.
Line 175-176. No wildlife reservoirs have ever been reported.
The reference cited says that many wild animals are infected. The specific species were added to the text (line 245)
A “systematic review” would entail a higher level of objectivity in the review of the literature without bias towards one particular conclusion. This review is not. 8 of 20 references cited were her own papers, while only including 4 papers (Citations 11-14) contradicted her hypothesis, totaling 3-4 sentences (line 45-52).
You are absolutely correct. Thank you for the suggestion to use “literature review.” This manuscript was not meant to be a systematic review and we have changed the opening line to use the word “literature review” instead. The only publications we are aware of that contradict the idea that BLV could infect humans and cause breast cancer, based on their own experiments, are already cited in this review (references #14 and #15). Please let us know of any other specific publications that are directly contradictory.
Dismissal #1:
Within this passage (Line 45-52), it was noted that the lack of detection of BLV via nest-generation sequencing does not “usually” integrate into the host genome (citing a review on retroviruses). This claim is unfounded for three reasons: (1) there is no evidence that this is the case for BLV (in cows or humans) and (2) NGS Library prep often does include PCR whole genome amplification, while bridge amplification of hybridized DNA fragments occurs during the sequencing run within the flow cell. (See page 5 Figure D Illumina’s Overview). (3) NGS sequencing is extremely sensitive in its ability to detect low abundant DNA molecules due to the massively parallel processes taking place leading to hundreds of millions of reads per run.
Thank you for this important information. We have omitted the entire paragraph we wrote about NGS.
Dismissal #2:
Within the Chinese study which demonstrated that no BLV seropositivity within Chinese women, it is noted that the kit used was indicated for bovine immunoglobulin, however there is more than sufficient evidence in the literature that secondary antibodies are polyclonal in origin and do typically cross-react to primary antibodies from multiple species, especially within mammals. Additionally, Reference 5 which shows that human sera is reactive to recombinant p24 via western blot, used protein G beads to conjugate HRP rather than secondary antibodies which directly contradicts (“cherry picks”) the logic used to dismiss claims made by Zhang et. al (Chinese antibody breast cancer paper; Citation [5])
The reason the use of the veterinary test kit seemed questionable to us for accuracy for human antibodies was that we had already tried using exactly the same test kit from IDEXX company, and all human serum samples we tested (about 80) were negative. We were very puzzled and disturbed, and called up the company to get more information. The lady who answered the phone connected me with the president and chief scientist of the company, John Lawrence. When I told him we used the kit to test human serum he said in a very annoyed voice, “DIDN’T YOU READ THE INSTRUCTIONS?!! The kit is intended for veterinary use only! The kit was designed to work with bovine serum and won’t work for human sera.” He then explained to me that the secondary antibody that comes with the kit is specific for the Fc portion of bovine immunoglobulins, and will not react with the Fc portion of the immunoglobulins of most other species, and in particular not with the Fc portion of human immunoglobulins. Therefore, the IDEXX kit was not a valid way to test human serum for antibodies to BLV.
Interpretation of Table 1:
At no point are the Sensitivity/specificity issues regarding FFPE in situ PCR and nested PCR included in the body of this manuscript as an objective discussion to qualify the findings summarized in Table 1. Both methods require stringent negative controls due to their proclivity to produce false positives. Additionally, the studies listed in support of this hypothesis (#6-8) cite the authors own work. To enumerate the claims and shortcomings of these papers:
[6] IS-PCR (figure 1), the negative control is “background staining”, excluding the presence of the primers used to “detect” BLV DNA. This negative control is not sufficient to conclude that positive staining is specific to BLV DNA due to the lack of evidence that these primers are not cross-reacting to formalin-fixed human DNA or simply produce artifactual staining.
For in situ PCR standard positive and negative controls are run: a BLV-cell line for the negative control and a BLV-positive cell line for the positive control. If these controls give the appropriate reaction then we can be sure that the PCT reactions on human tissues were valid. The “background control” for in situ PCR is an additional control for the in situ PCR method. Since the endpoint color reaction for BLV detection in the in situ PCR method is dark brown, we had to be sure that any dark-brown/black areas or dots in cells were not due to dirt, melanin from skin cells, or other artifacts of brown/black color in the same cells or tissue area. Therefore we called this special additional control the “background control.” It did not replace the standard negative control we used, a BLV-negative cell line. This concept and the need for the additional special control were explained in the original articles already published, but perhaps not adequately. We have added that to the text and do not feel that it was necessary to reexplain it in this review article.
It is not clear what the reviewer means by “figure 1” in the context of the negative controls. He or she is apparently referring to a Figure 1 in a previous publication where the meaning of a “background control” and the need for it were explained. In this review under consideration here, Figure 1 is quite different and consists of two maps of the world.
[7] IS-PCR and IS-hybridization (ISH) on FFPE (Figure 1). This experiment lacks controls (both positive, FLK cell line smear (mentioned) and negative, intact IS-PCR conditions, containing primers, on BLV-negative human tissue). When compared to ISH detection of HPV (Figure 3) These details of the how the negative control was performed (inclusion of ISH probe) are omitted. Given that this is an established method for HPV in cervical tissues, ISH probes were most likely included. Dr. Nuovo’s textbook was cited as the methods used for HVP ISH, not a specific primary publication.
See comment above in response to the previous critique [6]. It seems that the reviewer is reviewing a previous publication. See out response to comment [6} above, which also can be applied as a response to comment [7} about the “background control.”
[8] in situ-PCR using a hybridization probe on FFPE sections (Figure 1). This experiment is also missing the critical negative control which has the presence of the in-situ probe on BLV negative tissue. This is noted in the figure legend that the presence of the probe leads to “artifacts inherent in some tissues”.
In conclusion, the work summarized in this “systematic review” reflect more of a self-fulfilling scientific review of one viewpoint based on less-than robust studies. Contradictory logic is used to support the authors position. This paper is not really a comprehensive and objective summary of the connection of BLV and human breast cancer. The suggestion of causation is premature and unsubstantiated. For instance, listed in the abstract that up to 50% of all new US breast cancer cases may be prevented in the absence of BLV’s existence without mention or investigation into the mode of transmission or mechanism of carcinogenesis.
You are absolutely correct that the suggestion of causation is premature and unsubstantiated and it is extremely helpful to have someone else besides the authors review the text and give a reaction to it. This is very informative to understand how other people view the whole situation. Thank so much for spending so much time on this and doing such a thorough job.